# THE ROLE OF TASK COMPLEXITY IN EMERGENT ABILITIES OF SMALL LANGUAGE MODELS

## ABSTRACT

We investigate the relationship between task complexity and the minimum model size required for learning specific tasks in small transformer models. We focus on the ListOps dataset, consisting of nested math operations. We define the task complexity as the Kolmogorov complexity (KC) of the code solving the task, using a rough proxy for KC. We find a power-law relation between KC and parameters required to learn, suggesting number of parameters to learn harder task increases almost cubic in KC. On individual math operations, sum mod 10 is hardest to learn. Surprisingly, when combining tasks, we observe that sum is learned earlier and with fewer parameters when trained alongside max and median. Analyzing the model, we find strong evidence that models trained on sum alone and models trained jointly converge to different algorithms. Concretely, the sum alone model doesn't seem to have learned number properties in the embedding layer, likely memorizing the sum table. In contrast, models trained on three tasks (maximum, median and sum) reveals that joint training results in clear number-like properties. Finally, we also find evidence that the sum-only model utilizes its feedforward layer more than the jointly trained model. Conversely, the attention layer in the joint model is activated more than the sum model. Our findings suggest there is another dimension to emergent abilities in language models, namely the algorithms being learned, potentially impacting scaling laws.

## 1 INTRODUCTION

The scaling laws of language models have been a central focus in machine learning research, describing how performance varies with model size, computational cost, and dataset size (Hoffmann et al., 2022; Muennighoff et al., 2024; Hestness et al., 2017). However, these laws have largely overlooked a crucial factor: the intrinsic complexity of the tasks being learned. Our work addresses this gap by introducing a quantitative measure of task complexity and exploring its relationship with model size and emergent abilities.

Using the ListOps dataset (Nangia & Bowman, 2018) as our experimental framework, we investigate how the complexity of mathematical operations—specifically maximum (max), minimum (min), median (med), and sum modulo 10 (sum)—influences the learning dynamics of transformer models. By approximating Kolmogorov complexity (KC) (Kolmogorov, 1965; Li et al., 2008) through the compressed size of minimal Python code implementing these operations, we establish a novel, quantifiable metric for task complexity. Our study reveals several groundbreaking findings:

1. **Cubic model growth with complexity:** We discover a power-law relationship between task complexity (as measured by KC) and the minimum model size required for learning. This quantitative link between complexity and model capacity offers a new perspective on scaling laws, suggesting they are less universal than previously thought and are significantly influenced by task complexity.

2. **Change in Algorithmic Complexity:** Surprisingly, in multitask settings, we observe that the sum operation, when trained alongside max and median, is learned earlier and with fewer parameters than when trained in isolation. This finding challenges conventional wisdom about task difficulty and suggests that curriculum design and task diversity play critical roles in model learning.

3. **Smarter Solution via Joint Training:** Our analysis of embedding layers reveals that models trained on three tasks (max, med, sum) develop meaningful numerical representations. In contrast, models trained solely on sum rarely exhibit these structures, suggesting they rely more on memorization than understanding.

These results provide compelling evidence that joint training guides models towards finding alternative, more efficient solutions. While training on sum alone may lead small models to memorize sum tables, joint training appears to push the system towards learning good number embeddings. This, in turn, enables the model to discover more efficient algorithms for performing sum operations, rather than relying on rote memorization.

Our work bridges the gap between algorithmic complexity theory and practical neural network training while challenging our understanding of scaling laws. It highlights the importance of considering task complexity and training diversity when designing and deploying models. Gaining a deeper insight into how task complexity, model capacity, and learning interact is essential for building more effective systems.

## 2 METHODOLOGY

We investigate the relationship between task complexity and model learning dynamics using KC as a measure of task difficulty. We employ the ListOps dataset to study nested mathematical operations, focusing on maximum, minimum, median, and sum modulo 10. Our approach involves training small-scale transformer models and analyzing their performance and internal representations. In this section, we describe our analysis methodology, including our approximation of KC, the specifics of the ListOps dataset, our model architecture, and our evaluation techniques.

### 2.1 KOLMOGOROV COMPLEXITY AND ITS APPROXIMATION

Kolmogorov Complexity (KC), also known as algorithmic complexity, provides a formal measure of the information content of a string (Kolmogorov, 1965; Li et al., 2008). For a given string $s$, its KC $K(s)$ is defined as the length of the shortest program that produces $s$ as its output. While KC offers a powerful theoretical framework for quantifying complexity, it is fundamentally incomputable (Li et al., 2008), necessitating the use of approximations in practical applications.

In this study, we employ a proxy measure for KC based on the compressed size of minimal Python code implementing each task. This approach aligns with the intuition that more complex tasks require longer descriptions, and thus, larger compressed sizes. Our method leverages the Lempel-Ziv (LZ) compression algorithm (Ziv & Lempel, 1977), which forms the basis of the widely used gzip compression tool. The use of compression algorithms as proxies for KC is well-established in the literature. Cilibrasi & Vitanyi (2005) introduced the Normalized Compression Distance (NCD), which uses standard compression algorithms to approximate the ideal Normalized Information Distance based on KC. This approach has been successfully applied in various domains, including bioinformatics, musicology, and plagiarism detection (Li et al., 2004).

Our specific methodology of using gzip-compressed code size as a KC proxy follows the precedent set by several researchers. Zenil et al. (2014) employed a similar technique in their study of algorithmic complexity for short strings. In the context of machine learning, Dingle et al. (2018) used compressed code length as a complexity measure to study the learnability of Boolean functions. By applying gzip compression to minimal Python implementations of our target tasks, we obtain a practical, computable proxy for KC. This allows us to quantify task complexity and examine its relationship with the minimum model size required for successful learning.

It's important to note that while this proxy provides a useful approximation, it has limitations. The choice of programming language (Python in our case) and the specific implementation details can influence the compressed size. Additionally, for very short or highly regular strings, compression-based approximations may not fully capture the theoretical KC. To mitigate these issues, we establish a set of ground rules (Appendix C) to ensure we do not use advanced features that do not capture the complexity of the algorithm (e.g. we do not use addition or sorting). Despite these constraints, our approach offers a pragmatic and widely accepted method for estimating relative task complexities in the context of machine learning and emergent abilities in language models.

## 2.2 EXPERIMENTAL SETUP

**Choice of task**   For our study on the emergence of abilities in language models, we chose the ListOps dataset (Nangia & Bowman, 2018) as our primary experimental framework. This choice was motivated by several key factors that align with our research objectives:

1. **Procedural Generation:** ListOps is a synthetically generated dataset, allowing us to create a large volume of examples with controlled properties. This feature enables us to fine-tune the complexity and distribution of our training and evaluation data.

2. **Exact Evaluation:** The mathematical nature of ListOps operations ensures unambiguous, exact evaluation of model outputs. This precision is crucial for accurately assessing model performance and learning trajectories.

3. **Adjustable Complexity:** ListOps offers a framework to modulate problem complexity by combining different operations and adjusting nesting depths. This flexibility allows us to create a spectrum of task difficulties, enabling a nuanced study of the relationship between task complexity and model size.

4. **Inter-task Relationships:** The dataset's multiple operations (max, median, sum modulo 10) provide an opportunity to explore task synergies and interference. This aspect led to our key discovery regarding the benefits of joint training, particularly in accelerating the learning of more challenging operations like sum modulo 10.

**Dataset Description**   ListOps consists of nested mathematical equations involving operations such as min, max, median, and sum modulo 10 applied to single-digit numbers (0-9) (Appendix H.1). It uses the Polish notation: (operation, [inputs]) To disentangle any complexity arising from tokenization we further simplify these expression by representing them by symbols: '+' for max, '−' for min, '/' for median, and '%' for sum modulo 10. For example `(max,3,(min,7,4,9))=4` becomes `s(+3(-749))=4e`, with 's' and 'e' being start and end tokens.

**Tokenization and CoT:**   We employ a character-based tokenization strategy for processing ListOps expressions. (Appendix H.2). We find that directly solving nested ListOps in one step can be quite challenging for transformer model (Fig. 7) Even with a **maximum of three nesting levels with three operands (inputs)** we find that GPT models with over 10 million parameters still fail to learn the task. To enhance model performance, particularly on more complex operations like sum modulo 10, we introduced a chain of thought (CoT) approach in our training data (Appendix H.3): `s(%12(%34))>(%127)>0=0e`, wehre the '>' token means one step of CoT, wherein we solve the right-most, inner-most parenthesis.

**Performance Evaluation**   To assess model performance, we randomly select 1000 equations from a held-out test set. The model is prompted to generate solutions character-by-character, starting from the equation prompt (e.g., `s(+3(-749))>`). We use the output to produce two metrics

1. **Loss:** using cross-entropy computed for every character of the output.

2. **Accuracy:** or "number of correct" answers evaluated using *only the final answer*, which we define as the first character after the first '=' symbol.

We then calculate the percentage of correctly solved equations, providing a clear metric of the model's problem-solving accuracy.

**Model Architecture**   For our experiments, we employed a series of tiny GPT models, inspired by the nanoGPT architecture (Karpathy, 2022). These models were specifically designed to be minimal yet functional implementations of the GPT architecture, allowing us to explore the relationship between model size and task performance with fine-grained control. Each model in our study uses a single attention head. We set the feedforward hidden dimension to four times the embedding dimension, a common practice in transformer architectures, providing sufficient complexity in the feedforward networks while keeping the model size constrained. By varying the embedding size and the number of transformer layers, we can create a range of models with different parameter counts.

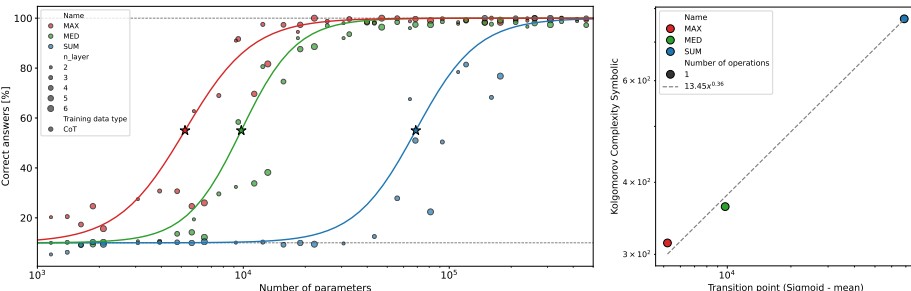

Figure 1: **Learning max, med and sum. Left:** As the number of parameters increases, the model demonstrates the ability to learn different operations. Max requires the fewest parameters to learn, while sum demands the most, reflecting the relative complexity of the task. We report the average results from five simulations for each configuration, fitting a sigmoid curve to the data. Transition points (Sigmoid - mean), marked by stars, indicate the midpoints of these fits. **Right:** The Kolgomrov Complexity, based on symbolic algorithms, vs the transition point. For the sum, the algorithm involves an explicit sum table and has therefore considerably higher complexity.

## 3 RESULTS

To understand how language models acquire mathematical abilities, we focus on accuracy as the primary metric for observing the emergence of learning, following the approach of Wei et al. (2022). While some subsequent studies have questioned the concept of emergence by examining other metrics (Schaeffer et al., 2024), we argue that this critique overlooks a fundamental principle from physics: during a phase transition, not all measurable quantities necessarily exhibit discontinuous changes. In a phase transition, only certain quantities, called the "order parameter", best captures the essential change in the system's state. For instance, in the transition from a paramagnet to a ferromagnet, the net magnetization serves as the order parameter, showing a sharp change at the critical temperature, while other properties may vary smoothly.

**Emergence of learning on single math operations.**   Based on the above, we believe that accuracy serves as the most appropriate order parameter for observing the emergence of mathematical abilities. While other metrics like loss may appear continuous (Fig. 13), accuracy often exhibits a more abrupt change as the model transitions from inability to ability in solving mathematical tasks. We first experimented with three individual operations in ListOps: max, med, and sum. We use equations with a maximum of three nesting levels and three operands per parenthetical group. Figure 1 left illustrates our findings from these basic experiments. As evident from the results, the difficulty of learning these operations varies significantly, with max being the easiest, then med. The sum operation, however, presents a substantially greater challenge, requiring a much larger number of parameters. This stark difference in learning difficulty among seemingly similar mathematical operations raises intriguing questions about the nature of task complexity and its relationship to model size.

### 3.1 AMBIGUITIES IN KOLMOGOROV COMPLEXITY

We now turn to Kolmogorov Complexity (KC) as a measure of algorithmic difficulty. The application of KC to our specific problem domain presents several ambiguities that require careful consideration.

**Multiple Solution Algorithms**   For each of the operations we study (max, med, sum), there exist multiple algorithms that can solve the problem. This multiplicity raises a critical question: *Which algorithm does our model actually learn?* The answer to this question has significant implications for our understanding of the model's capabilities and the nature of its learning process.

**Symbolic vs. Numerical Understanding**   To address this ambiguity, we consider two broad categories of potential solution algorithms: **1. Purely Symbolic Algorithms 2. Number-Based Algorithms**. This categorization allows us to explore whether the model develops a true understanding of numerical properties or merely learns to manipulate symbols without grasping their numerical significance. We define a set of ground rules for how to design symbolic or number based algorithms (Appendix C). Note that the number algorithm is still assumed to not use a simple python add

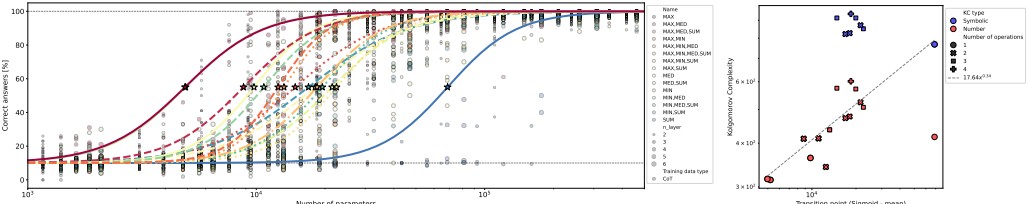

Figure 2: **Learning All Combinations of Operations. Left:** A sigmoid function was fitted to the data, with each configuration run across five simulations. The midpoint of each sigmoid curve is marked by a star. Single operations use a solid line (-), double operations are dashed (- -), triples use a dotted line (:), and all four operations are marked with dot-dash lines (:.). **Kolmogorov Complexity.** For symbolic KC, we use a lookup table for sum and a sorted character list for min, max, and med. For numerical KC for sum is implemented using bit-wise operations. **Right:** Comparing symbolic (blue) and numeric (red) KC as functions of the transition point. The primary difference between these metrics lies in operations involving sum, where the symbolic KC is significantly higher due to the lookup table.

operation. Rather, it needs to implement that from basic boolean operations. Our rules are designed to mimic potential learning strategies of the model rather than to define all possible algorithms.

**Model Inference and Algorithmic Complexity** We posit that a naively trained model may not develop a deep understanding that the symbols represent numbers. Consequently, it is more likely that the model learns a symbolic algorithm. However, at this stage, this remains a hypothesis that requires further investigation. To explore this hypothesis, we calculate the KC for both symbolic and number-based algorithms for each operation. Yet, even for symbolic solutions, there are multiple choices: Does the model form an array, sort the elements, conditionals, or all of the above? While designing solutions for max and med, we found the most straightforward symbolic solutions involved keeping a sorted array of symbols and conditionals for comparing order of appearance of elements. For med, it also involves a brute-force sorting algorithm, again only using conditionals. Yet, these algorithms would not look significantly different when implemented with numbers, the only difference being that the array of symbols does not need to be kept. Therefore, the difference between our numeric and symbolic solutions for max and med is very small.

Another naive solution for max would be to memorize the full table of comparisons, but we think this is algorithm is unlikely to be learned in our experiments for multiple reasons. First, the table requires large memory, yet even very small models learn max. Second, we show below that max model quickly learn an embedding of numbers that is sorted, strongly suggested an algorithm based on a sorted list is being learned. For the sum operation, however, we believe a symbolic solution without using number properties would require learning the full sum table, meaning an array of all tuples $[a, b, (a + b) \bmod 10]$. Based on this, to attach a KC value to the transition point in Figure 1 left, we choose symbolic solutions based on sorted lists for max and med, and based on memorizing the sum table for the sum.

Figure 1 right demonstrates the complexity of the max, med, and sum operations plotted against the learning transition point $n_p$ (number of parameters at which learning occurs). Interestingly, we observe an almost linear relationship in this log-log plot, suggesting a power-law relationship, $KC \approx 13.5 n_p^{0.34}$. In other words, learning grows almost as $KC^3$. This observed power-law relationship raises an intriguing question: Does this trend continue for more complex tasks? To explore this further, we considered whether combining these operations might increase the overall task complexity and provide additional data points along our KC-transition fit. This approach not only allows us to test the robustness of our observed power-law relationship but also provides insight into how models handle multi-operation tasks. Moreover, these experiments set the stage for a deeper investigation into what the model actually learns, paving the way for our subsequent analyses of embedding layers and attention mechanisms.

## 3.2 JOINT TRAINING ON MULTIPLE TASKS

To further explore the relationship between task complexity and model learning, we expanded our investigation to include all possible combinations of four operations: max, min, med, and sum. The

results, illustrated in Figure 2 reveal a surprising and counterintuitive pattern. Contrary to our initial expectations, we observed that models trained on mixed tasks require fewer parameters to achieve proficiency compared to those trained on the sum operation alone. MAX and MIN achieve 100% accuracy first, followed by the MAX+MIN+MED combination, and then all other combinations involving sum, with the single 'sum' operation being learned last. This finding challenges the intuitive notion that increasing task complexity (by combining multiple operations) would necessitate larger models or more extensive training.

Interestingly, the relationship between the number of operations and the transition point (where the model begins to learn effectively) is not straightforward. Our analysis reveals no strong correlation between these factors, suggesting that the mere quantity of operations is not a reliable predictor of learning difficulty. Furthermore, when we attempt to correlate these results with a naive symbolic KC estimate (Figure 2, right, blue dots), we find that the previously observed power-law relationship appears to break down. Additionally, if we use KC based on numeric algorithms (red dots), we get a different pattern, which is more along the previous power-law, except for sum.

These observations present us with a significant puzzle: What factors determine the unexpected shifts in transition points for joint tasks? The fact that sum is learned with fewer parameters in mixed operation settings strongly suggests that joint models are discovering more efficient algorithms than those learned when training on sum alone. This leads us to several critical questions:

1. What is the nature of these more efficient algorithms?

2. Are these algorithms leveraging numerical properties, or are they still primarily symbolic?

3. How can we gather evidence to understand the underlying mechanisms at play?

To address these questions and gain deeper insights into what the models are actually learning, we turn to more sophisticated analysis techniques. In the following sections, we will explore the use of Principal Component Analysis (PCA) on the models' embedding layers. Additionally, we will employ various probing techniques to examine the roles and behaviors of different model components during the processing of mixed operations.

### 3.3 EMBEDDING ANALYSIS: UNCOVERING NUMBER REPRESENTATIONS

The original "grokking" studies, such as Power et al. (2022), typically focused on binary operations with pairs of symbols as input. These studies revealed that after the model finally learned the operation (e.g., modular addition), the embeddings of the input symbols suddenly organized into a circular structure, visible after dimensionality reduction techniques like t-SNE. Subsequent work, such as Liu et al. (2022), demonstrated that this circular structure could be observed directly through Principal Component Analysis (PCA).

Inspired by these findings, we examined the embedding layers of our models trained on the ListOps dataset. We analyzed both the cosine similarity matrix of the embeddings and their principal components. Figure 3, top, shows this analysis for a model trained solely on the sum operation, while the second row presents the same analysis for a model trained on all three operations (max, med, sum), which we refer to as the "all3" model.

**sum-only Model** For the sum-only model (Fig. 3), which achieved 96%, we observe:

1. The cosine similarity of the embedding weights shows no clear pattern among numbers.

2. The first three principal components (PC1, PC2, PC3) of the number embeddings display no discernible structure when plotted against each other.

These observations suggest that despite achieving high accuracy, the sum-only model has not developed a structured representation of numbers.

**All3 Model** In contrast, the All3 model (Fig. 3) exhibits remarkable structure in its embeddings:

1. The cosine similarity matrix reveals a clear band of strong correlation around the diagonal, indicating that consecutive numbers have correlated embeddings.

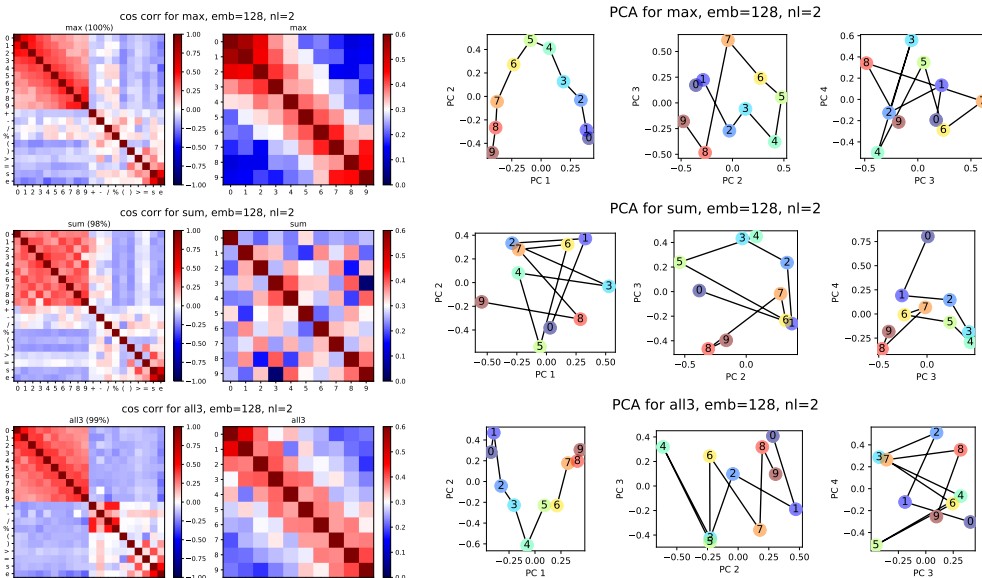

Figure 3: **Cosine Correlation and PCA:** We show three models trained on: max (top row), sum (middle), and all3 (max,med,sum, bottom row). The accuracy of each model is shown in the title of the first column (all above 98%). Left-most column shows cosine correlation matrix for all symbols (see axis labels). While the numbers (first red block) have generally high correlations, their correlation patterns differ in each model. The second column zooms on the number block of cosine correlation. In max and all3 we see a clear band of strong correlation around the diagonal. The last three columns show PCA plots of $PC_i$ vs $PC_{i+1}$. In max and all3, we see that PC1 sort the numbers, While PC1 and PC2 form an arc together. Additionally, in all3 PC3 seems to separate odd from even numbers.

2. PC1 almost perfectly sorts the numbers from 0 to 9, suggesting a learned ordering of numerical symbols.

3. PC1 vs. PC2 forms a distinct arc, reminiscent of the circular structures observed in grokking studies, but with an open side due to PC1's sorting behavior.

4. Intriguingly, PC3 displays a zig-zag pattern that perfectly separates odd from even numbers.

These patterns provide strong evidence that the all3 model has learned multiple number properties, including ordering and parity. The emergence of the odd-even distinction is particularly noteworthy, as it's not strictly necessary for any of the three operations. This could indicate that the model has developed a more generalized understanding of number properties beyond what's directly required for the tasks.

**Implications and Hypotheses** The stark difference between the sum-only and all3 models leads us to several hypotheses:

1. **The sum-only model**, despite its high accuracy, **may be primarily memorizing** the sum table rather than learning generalizable number properties. This could explain its high transition point in terms of required parameters.

2. **The all3 model**, through exposure to diverse operations, appears to have developed a more nuanced and generalizable representation of numbers. This richer representation might enable more efficient algorithms for all operations, including sum (e.g. bitwise).

3. **The emergence of number properties** not strictly necessary for the tasks (e.g., parity) suggests that joint training on diverse operations promotes a more comprehensive understanding of the underlying domain.

4. **The circular-like structure** in the all3 model's embeddings, similar to those observed in grokking studies, might indicate the model is learning a number representation similar to the grokking studies.

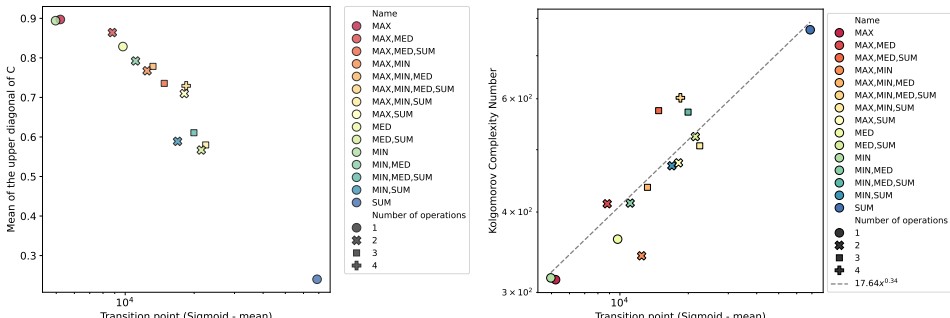

Figure 4: **Criteria for Symbolic vs Numeric KC. Left:** The mean of the two upper diagonal bands of the learned cosine similarity matrix $C$ of the embeddings of number symbols vs transition point. Efficiently learned operation combinations display a high mean, suggesting that the model may have internalized number characters and employed a number-based algorithm. The only outlier is the sum, which shows no number properties, implying that the model may be learning a symbolic solution, such as a lookup table. **Right:** Plotting numeric KC for all combinations except sum, and symbolic KC for sum, we recoveralmost exactly the same power-law discovered for single operations.

These findings provide a potential explanation for the counterintuitive results observed in our transition point analysis. The all3 model's lower transition point for learning sum, compared to the sum-only model, may be attributed to its more efficient, number-property-based algorithm rather than rote memorization. In the following sections, we will further probe these models to validate these hypotheses and explore the implications for efficient algorithm learning in neural networks.

## 3.4 REFINING COMPLEXITY ESTIMATES: FROM SYMBOLIC TO NUMERICAL ALGORITHMS

Building upon our observations from the embedding analysis, we hypothesized that the cosine correlation matrix of the embedding layer could serve as an indicator of whether the model learned a number-based algorithm or a purely symbolic one. This hypothesis, while not definitive proof, is based on the striking differences observed between the sum-only and all3 models in their embedding structures.

**Cosine Correlation as a Proxy for Algorithm Type** To test this hypothesis, we plotted the mean of the cosine correlation matrix against the transition point for each model configuration (Figure 4 left). Remarkably, we observed an almost perfect correlation in this semi-log plot. This strong relationship suggests that the degree of structure in the embedding space, as measured by the average cosine similarity between embeddings, is closely tied to the model's learning efficiency. The plot reveals a clear trend: models with higher mean cosine correlation (indicating more structured number representations) tend to have earlier transition points. This observation aligns with our earlier hypothesis that models learning number-based algorithms, rather than purely symbolic ones, achieve proficiency with fewer parameters. Additionally, we observe a large gap between the sum-only models and the rest, suggesting that the sum-only model may be the sole model that did not develop any understanding of the numbers.

**Revising Kolmogorov Complexity Estimates** Based on these findings, we revised our approach to estimating KC for tasks involving the sum operation. For models showing high correlation between embeddings of consecutive numbers (everything except sum-only), we now assume that some numerical characteristics were learned. Consequently, we switched from a symbolic algorithm to a number-based algorithm for calculating KC in these cases. Specifically, instead of using a simple lookup table or symbolic manipulation for the sum operation, we implemented a bit-wise sum algorithm. This approach more closely aligns with the kind of numerical understanding we believe the model has developed, based on the embedding analysis.

**Improved KC vs. Transition Point Relationship** The result of this revised KC calculation is a dramatically improved relationship between KC and the transition point (Figure 4 right). With

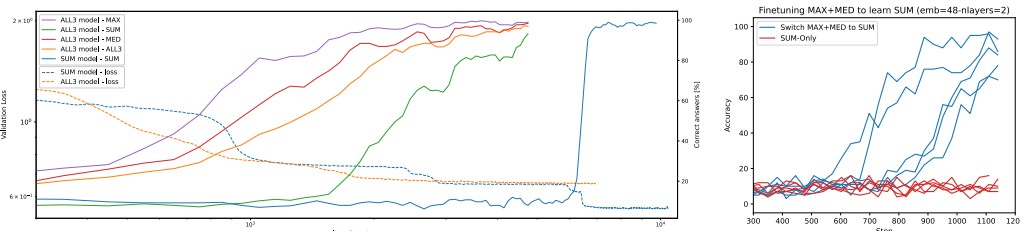

Figure 5: **Left: Evolution of loss and accuracy in all3 and sum models.** Both models have 128 embedding dimension. The all3 model, evaluated on test sets for max, med, and sum operations individually. The sum model, trained and tested solely on sum data, demonstrated slower learning of the sum operation compared to the all3 model. Interestingly, the all3 model showed significantly earlier learning of the sum operation, suggesting that the presence of max and med operations during training may provide beneficial signals that accelerate the learning of sum. **Right: Learning sum by finetuning max,med:** We train model much smaller than the sum-only learning transition (48 embedding). By switching the training data slowly from max+med to pure sum (never showing expression mixing all three) the model is able to learn sum (blue) in much this low parameter regime. In comparison, sum-only models (red) did not learn at this size.

this adjustment, we observe that all data points, regardless of the specific operation or combination of operations, now follow approximately the same power-law curve. This refined plot reveals a striking pattern: the relationship between task complexity (as measured by our revised KC) and the model size required for learning (transition point) follows a consistent power law across all task combinations. The fitted curve, given by $KC \approx 18n_p^{0.34}$, suggests a fundamental scaling relationship between task complexity and required model capacity. It predicts a cubic growth in model size with task complexity. Further studies would be needed to see if this relation extends to other datasets.

## 3.5 Temporal Dynamics of Learning and Transfer

To further investigate our hypotheses about the nature of learning in single-task versus multi-task settings, we conducted a series of experiments tracking the accuracy of models throughout their training process. These experiments not only corroborated our earlier findings but also revealed intriguing dynamics in how models acquire and transfer knowledge across tasks.

**Rapid Multi-task Learning vs. Delayed Single-task Learning** We trained twp models, one on sum alone and one all3, at 128 embedding size, which is above transition point for both. The results, illustrated in Figure 5, reveal striking differences in learning dynamics:

1. The all3 model demonstrated rapid acquisition of all three operations (max, med, sum) within approximately 3000 training steps.

2. In contrast, the sum-only model exhibited a significant delay before showing any signs of learning, with the lag varying across multiple runs.

Moreover, when evaluating the all3 model on individual tasks, we observed a subtle but important sequence in learning: MAX was learned fastest, followed closely by MED, and then sum. Crucially, the gaps between the learning of these operations were minimal, with the model beginning to grasp sum while still perfecting MAX and MED.

These observations strongly support our hypothesis that the development of a robust, number-like internal representation plays a pivotal role in facilitating rapid and efficient learning across multiple operations.

**Transfer Learning and Representation Stability** To further explore this hypothesis, we designed a transfer learning experiment: 1) We first trained a model (embedding size 48) on max and med operations until proficiency. 2) We then gradually introduced sum operations, increasing their proportion in the training data from 0% to 100% over 1000 steps, while simultaneously phasing out max and med.

Key findings from this experiment include:

- The model began learning sum immediately upon its introduction, despite never seeing mixed expressions (e.g., sum with max or med).
- Interestingly, the model started to forget max and med once these operations were no longer present in the training data.
- Crucially, we verified that the model retained its number-like embedding structure even after max and med were completely phased out.

Perhaps most strikingly, we found that even much smaller models (embedding size 24) could learn sum perfectly using this hybrid approach, mirroring the efficiency of all3 models. This resulted in a model capable of performing sum operations that was 7x smaller than the vanilla sum-only model, while relying on a more sophisticated, number-like internal representation.

**Attention and feedforward layers.** To investigate the internal dynamics of the sum-only vs all3 models, we focused on the final layer of a 3-layer transformer network, featuring a single attention head and an embedding dimension of 128 (Appendix I. We find:

1. In the attention sublayer, the 'all3' model shows slightly higher ratio values, **suggesting that attention mechanisms play a more pronounced role when the model is trained on diverse operations.** This could indicate that the attention sublayer is capturing more complex patterns or relationships necessary for handling multiple operations.

2. Conversely, in the feedforward sublayer, the 'sum' model demonstrates significantly higher ratio values. This **suggests that when trained on 'sum' alone, the model relies more heavily on the feedforward network for computation.** This could imply that the 'sum' operation is being implemented more directly through feedforward transformations.

3. The larger effect size in the feedforward layer (-0.3379) compared to the attention layer (0.1761) indicates that the difference in behavior is more pronounced in the feedforward component.

These observations suggest a trade-off in how the network allocates its computational resources. The 'all3' model appears to leverage its attention mechanism more, potentially to handle the diversity of operations it was trained on. In contrast, the 'sum' model seems to channel more of its computation through the feedforward network, possibly developing a more specialized but less flexible approach to solving the sum operation. We discuss further implications in Appendix J.

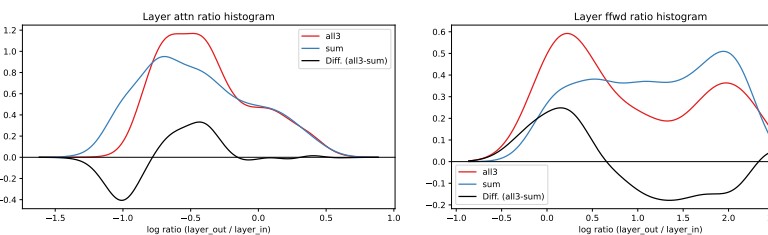

Figure 6: **Comparison of layer output/input ratio distributions** for models trained on all three operations (all3) versus sum operation alone (sum). **Left:** Attention layer ratio histogram. **Right:** Feedforward layer ratio histogram. The x-axis represents the log ratio of layer output norm to input norm, while the y-axis shows the density. The black line represents the difference between the all3 and sum distributions (all3 - sum).

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

## A    Related works

The scaling laws of language models describe how performance, measured by validation loss, varies with model size (number of parameters) Hoffmann et al. (2022), computational cost (number of FLOPs) Muennighoff et al. (2024), training dataset size (number of data point) Hestness et al. (2017), and knowledge storage capacity (in bits) Allen-Zhu & Li (2024), providing a guideline for designing new models. However, little is known about how the model size scales with the task complexity. There is no consensus on how to measure task complexity in language tasks, and even simple mathematical operations lack clear complexity metrics.

In general language models performs poorly on symbolic mathematical tasks Frieder et al. (2024); Dziri et al. (2024); Dave et al. (2024) such as the ListOps dataset Nangia & Bowman (2018) used in this study. Models often struggle with generalization, tending to memorize tasks rather than simulate the underlying algorithms, akin to using look-up tables. While mathematical tasks prove challenging for large language models to learn, they provide a controllable playground to test how models learn different tasks and to evaluate their accuracy quantitatively. The complexity of a mathematical task has been defined previously in various ways, such as the number of bits needed to memorize the task Dave et al. (2024), the number of operands and nesting depth Petruzzellis et al. (2024), and the computational graph Dziri et al. (2024). However, these metrics do not capture the true complexity of the underlying task. Furthermore, it is difficult to determine the exact algorithm learned by the model.

Here, we analyze how language models learn mathematical tasks by training small models on the ListOps dataset. The ListOps dataset allows us to measure model accuracy explicitly by asking the model to solve mathematical equations. Using this bottom-up approach, we show that once the number of trainable parameters in the model reaches a critical point, the model is able to learn the task. When training models on datasets based on different levels of operation difficulty, this critical point shifts according to the complexity of the task. Contradictorily, we find that combining operations makes the task easier for the model than individual operations, suggesting that task diversity plays a significant role.

## B    Designing solution algorithms

## C    Ground Rules for Algorithm Design

To systematically analyze these possibilities, we establish two sets of ground rules for designing algorithms in each category:

**Rules for Purely Symbolic Algorithms:**

- Use of if, else, and or conditions for comparing symbols is allowed
- Function definitions and calls are permitted
- For loops and lists of symbols can be used
- Comparing positions in a list is allowed (as it can be implemented with loops and conditionals)
- No mathematical operations or direct comparison of numbers are allowed

**Rules for Number-Based Algorithms:**

- Direct mathematical operations (addition, subtraction, etc.) are not allowed
- Binary or representation and boolean bitwise operations are permitted
- All operations allowed in the symbolic case are also allowed here

### C.1    Example Algorithms

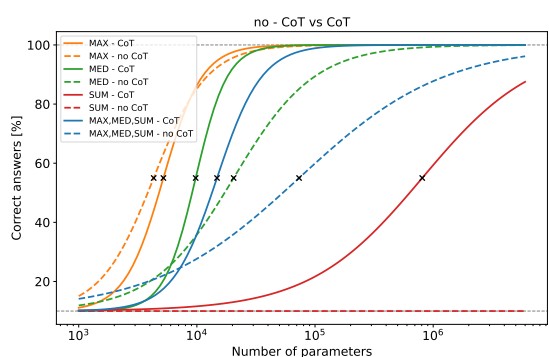

Figure 7: **no - CoT vs. CoT.**

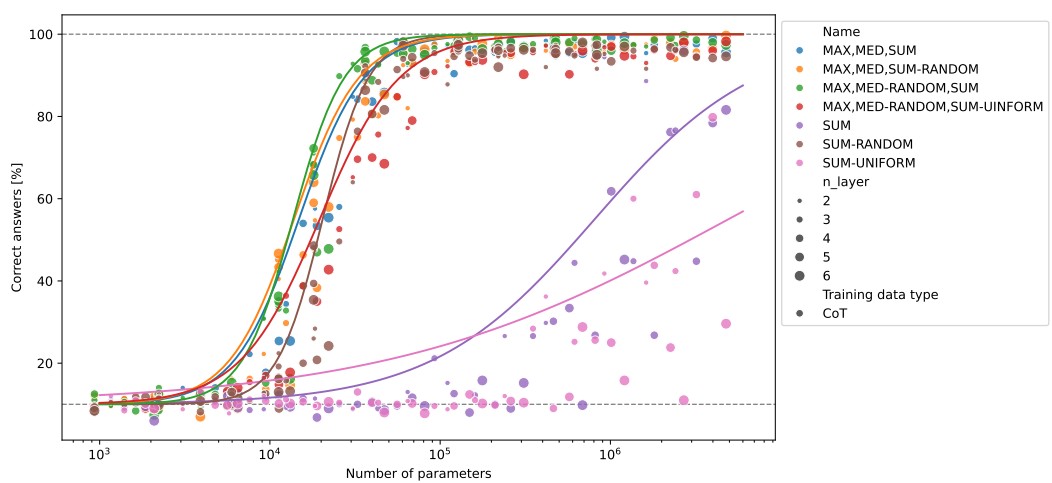

Figure 8: **Random sum table.**

# D COMPLEXITY: NUMBER OF OPERANDS AND NESTING LEVEL.

In previous studies, task complexity has been characterized through various measures, including the number of bits required to memorize the task, which corresponds to the length of the expression Dave et al. (2024), the number of operands and nesting depth Petruzzellis et al. (2024), and the structure of the computational graph Dziri et al. (2024), which captures the number of nestings. Here, we examine how the number of operands and nesting levels affect the learning ability of small language models. Focusing on the `all3` task, which combines `max`, `med`, and `sum` operations, we manipulate complexity by varying the number of operands (`arg = {3, 4, 5}`) and nesting depth (`depth = {3, 4, 5}`). The length of equations, measured by the number of characters, depends on both the number of operands and the nesting level. We find that nesting has a greater impact: increasing the nesting level from 3 to 4 results in longer equations than increasing the number of operands from 3 to 5 at a fixed nesting level (Fig. 9). We train the GPT model on the `all3` dataset with all combinations of `arg` (number of operands) and `depth` (nesting levels), finding that the model's performance correlates with the sum of operands and nesting levels (`arg + depth`). Notably, transition points tend to group together for configurations with the same sum (Fig. 10, 11b). While Fig. 11a demonstrates that the model requires more parameters to solve longer equations, it also indicates that `arg + depth` serves as a reliable predictor of the transition point. We also calculate the total complexity for each equation, which is a combined metric $C_{total} = wC_{Character} + wC_{Nesting} + wC_{Branching} + wC_{Diversity} + wC_{OperationComplexity}$ (Fig. 19), showing strong correlation with equation length. In calculating $C_{total}$ only the $C_{OperationComplexity}$ term was fixed to the `all3` operation, demonstrating that the model's ability to learn the task depends on the number of operands and the nesting level (Fig. 10). However, the operation complexity, defined by $C_{OperationComplexity}$ represents an independent challenge and follows a different learning pattern, as shown in the main paper.

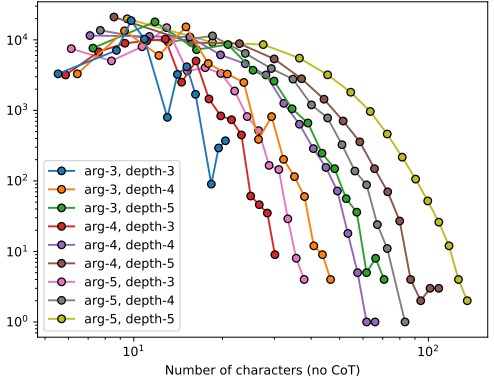 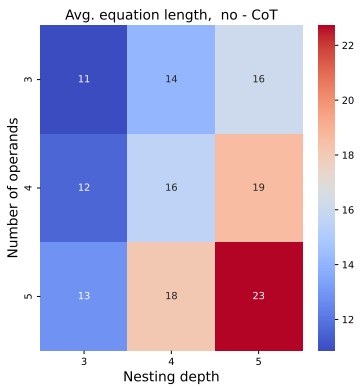

(a) **all3: Distribution of equations length.**      (b) **all3: Average equations length.**

Figure 9: **Equation length.** Fixing the task to all3 and varying the number of operands (arg) and nesting levels (depth), we generate 50,000 equations. (a) Distribution of equation lengths without solution steps (no CoT), showing that equation length increases more rapidly with nesting level than with the number of operands. (b) Heat map of average equation length as a function of nesting level and number of operations.

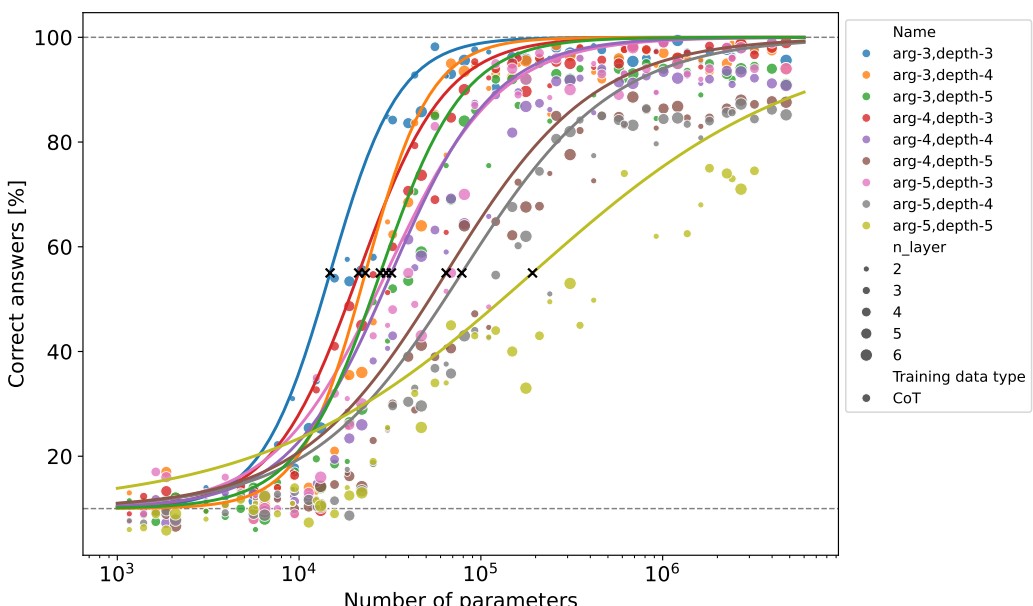

Figure 10: **Learning all3 operations with varying numbers of operands and nesting levels.** The model requires more parameters as the number of operands and nesting levels increases. Higher nesting levels particularly demand larger model sizes to learn the task. We present the average of five simulations for each configuration and fit a sigmoid function, with the cross marking the middle value (transition point). The transition points reveal an interesting pattern: the sum of the number of operands and nesting levels groups together. For example, the transition points for arg-3,depth-4 (orange) and arg-4,depth-3 (red) are close to each other, as are those for arg-4,depth-5 (brown), and arg-5,depth-4 (grey).

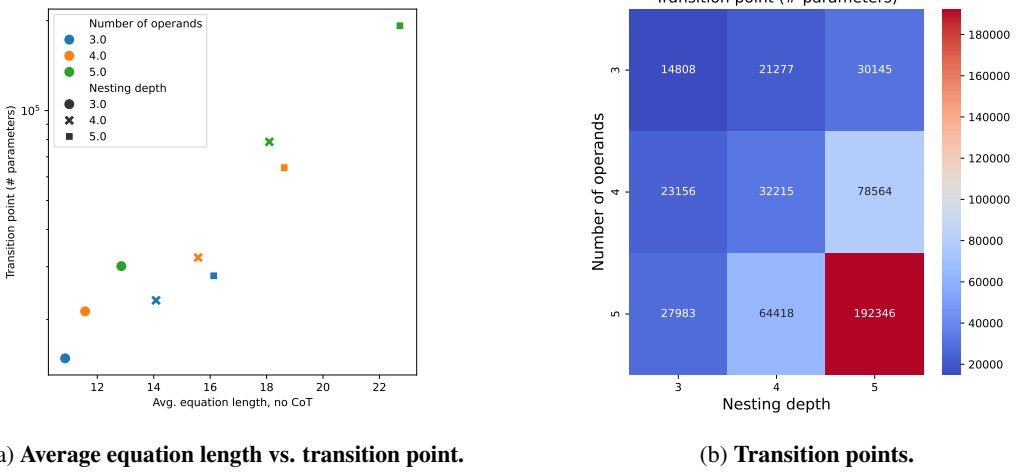

(a) **Average equation length vs. transition point.**                    (b) **Transition points.**

Figure 11: **Transition point.** (a) Transition point in function of the average equation length. (b) Heat plot of transition point in function number of operands and nesting depth.

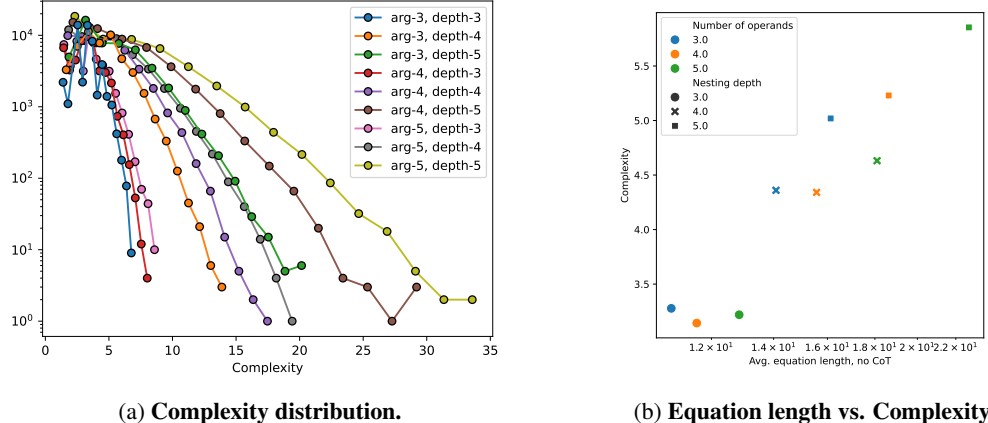

(a) **Complexity distribution.**  (b) **Equation length vs. Complexity.**

Figure 12: **Combined complexity.** (a) Complexity distribution. Increasing depth results in increasing complexity. (b) Complexity in the function of the average equation length.

# E   EXPERIMENT SETTINGS.

1. Model: GPT model (Karpathy (2022))

2. Number of layers: 1, 2, 3, 4, 5, 6

3. Number of head: 1

4. Embedding dimension: 4, 8, 12, 16, 24, 32, 48, 64, 96, 128, 192, 256

5. Number of random seed: 5

6. Context window size: 128

7. Batch size: 128

8. Learning rate: 5e-4

9. Optimizer: Adam

10. Early Stopping criteria: minimum change in the average of the last 100 steps of the training loss value $\Delta_{min}$ = 2.5e-3 with 10 patience steps or increasing validation loss, checking after 2000 steps

11. Training data size: 50k equations (first 90% training data, the last 10% validation data, test data sampled from the validation data set)

12. Fixed character map (19 tokens): s, e, (, ), >, +, -, %, /, 0, 1, 2, 3, 4, 5, 6, 7, 8, 9

# F   EXPERIMENTS ON LISTOPS.

1. no CoT/CoT: MAX, MED, SUM, MAX/MED/SUM

2. all combination: MAX, MIN, MED, SUM

3. random operation tables: MAX/MED/SUM-RANDOM, MAX/MED-RANDOM/SUM-UNIFORM-MAP, SUM-RANDOM, SUM-UNIFORM-MAP, PRODUCT

4. training size: MAX/MED/SUM with 2 layers and 128 embedding dimensions

5. training data complexity: MAX/MED/SUM, all combination of 3,4,5 maximum nesting level and 3,4,5 maximum number of operands

6. context window size: MAX/MED/SUM with 2 layers and 128 hidden dimensions for a maximum of 3 nesting levels + 3 operands, and a maximum of 5 nesting levels + 5 operands

7. training accuracy: MAX, MED, SUM, MAX/MED/SUM 1,2,3 layers and 128 embedding dimensions - model saved each 100 steps.

# G EXPERIMENTAL SETUP

In our experiments, we employed a series of tiny GPT models inspired by the nanoGPT architecture (Karpathy, 2022). These models were specifically designed to be minimal yet functional implementations of the GPT architecture, allowing us to explore the relationship between model size and task performance with fine-grained control. Each model in our study uses a single attention head. We set the feedforward hidden dimension to four times the embedding dimension. This 4:1 ratio is a common practice in transformer architectures, providing sufficient complexity in the feedforward networks while keeping the model size constrained. By varying the embedding dimension and the number of transformer layers, we were able to create a range of models with different parameter counts. This approach of using scaled-down GPT models enabled us to conduct a detailed analysis of emergent abilities and multitask learning dynamics in a computationally efficient manner

In the scaling experiments, we consider three ListOps datasets: one whose equations consist of MAX/MED/SUM operations, one whose equations consist of MAX/MED operations, and one whose equations contain only only the SUM operation. All 3 datasets consist of 51k unique CoT examples, with-holding 1k samples as the test dataset on which we evaluate the model's generation accuracy. The remaining 50k samples were partitioned into a training set consisting of 45k (90%) samples, and a validation set consisting of 5k (10%) samples.

We sweep across GPT models the size of our transformer across embedding sizes of '[256, 192, 128, 96, 64, 48, 32, 16, 8, 4]' and layers '[2,3,4,5,6]', allowing a diverse range of models with different parameter counts. Each experimental configuration is repeated 5 times using different random seeds. Across all experiments, MLP-hidden-size:embed-size ratio is set to 4:1 and a single attention head is used. Each model is trained for 50k training steps with no dropout. Learning rate starts at 1e-4 and is exponentially decayed to 1e-5 over the course of training. Gradients are clipped to be norm 1, and we use the ADAM optimizer with weight_decay set to 1e-2. We use a mini-batch size of 64 each having 128 tokens which is the maximum context window for the models.

Each experiment used a single GPU on a machine consisting of 8x NVIDIA A40s. To accelerate the training speed of these experiments, we re-implemented the nanoGPT repository in JAX Bradbury et al. (2018).

To support our hypothesis that the MAX/MED operations assist smaller models to learn the more complex SUM operation, we fine-tuned models that were initially trained on MAX or MED operations on the SUM operation. These results are reported in . Experimental details are described below.

We consider only shallow 2-layer models configured at embedding sizes '[128, 32, 24]'. In this experiment, we use four ListOps CoT generated datasets, each consisting of 26k samples (after 1k of these is withheld as test data to evaluate generation accuracy, 90% is used as training and 10% is used as validation as before):

1. **MAX-only operations**

2. **MED-only operations**

3. **MAX/MED operations**

4. **SUM-only operations** (Baseline)

For all non-baseline experiments, an early stopping criteria is established that flags when validation loss starts increasing and/or the training loss stops decreasing. At this point, we slowly introduce SUM-only examples into each mini-batch containing SUM-only operations over 2000 training iterations, after which only SUM operations are used to train the model. For the baseline experiment, we train on examples containing only the SUM operation throughout training.

Each experimental configuration is repeated for 5 different random seeds. Checkpoints are saved and evaluated every 1k steps throughout training.

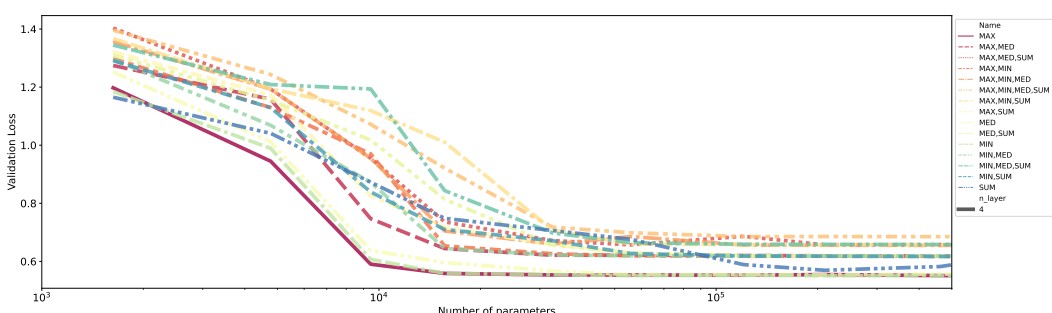

Figure 13: **Number of parameters vs Validation loss.** This plot illustrates the validation loss across all possible configurations with four layers. The curves converge at distinct points, corresponding to the number of operations being learned, with a smooth transition observed as the number of parameters increases.

## H DATA AND PROCESSING

### H.1 DATASET NOTATION

ListOps consists of nested mathematical equations involving operations such as min, max, median, and sum modulo 10 applied to single-digit numbers (0-9). It uses the Polish notation: (operation, [inputs]) For example: `max(3,min(7,4,9))=4`.

$$\texttt{max(3,min(7,4,9))=4} \quad \Rightarrow \quad \textbf{Polish: } \texttt{(max,3,(min,7,4,9))=4}$$

To disentangle any complexity arising from tokenization we further simplify these expression by representing the by symbols: '+' for max, '−' for min, '/' for median, and '%' for sum modulo 10. For example:

$$\texttt{(max,3,(min,7,4,9))=4} \quad \Rightarrow \quad \textbf{Our notation: } \texttt{s(+3(-749))=4e}$$

In this notation, 's' denotes the start of the expression, 'e' marks the end, and parentheses indicate nesting levels.

### H.2 TOKENIZATION

We employ a character-based tokenization strategy for processing ListOps expressions. This approach offers several advantages:

1. **Simplicity:** Character-level tokenization eliminates the need for complex tokenization rules or a large vocabulary.

2. **Generalizability:** It allows the model to potentially generalize to unseen number combinations or deeper nesting levels.

3. **Alignment with Task Structure:** The character-by-character nature of the tokenization matches the step-by-step problem-solving process we aim to induce in the model.

Each character in the ListOps expression, including digits, operation symbols, and structural elements (parentheses, 's', 'e'), is treated as a separate token. This granular representation enables the model to learn the syntactic structure of the expressions alongside their semantic content.

### H.3 CHAIN OF THOUGHT IMPLEMENTATION

We find that directly solving nested ListOps in one step can be quite challenging for transformer model (Fig. 7) Even with a **maximum of three nesting levels with three operands (inputs)** we find that GPT models with over 10 million parameters still fail to learn the task. To enhance model performance, particularly on more complex operations like sum modulo 10, we introduced a chain of

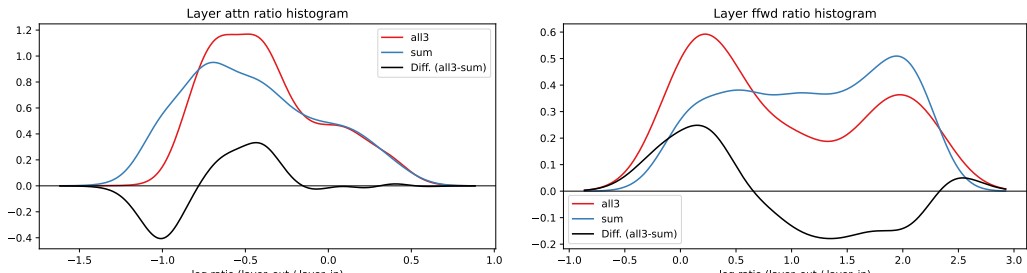

Figure 14: **Comparison of layer output/input ratio distributions** for models trained on all three operations (all3) versus sum operation alone (sum). **Left:** Attention layer ratio histogram. **Right:** Feedforward layer ratio histogram. The x-axis represents the log ratio of layer output norm to input norm, while the y-axis shows the density. The black line represents the difference between the all3 and sum distributions (all3 - sum). These plots illustrate distinct operational patterns between the two models, with the attention layer showing increased activity in the all3 model and the feedforward layer demonstrating higher ratios in the sum model.

thought (CoT) approach in our training data. This method involves providing step-by-step solutions that resolve the deepest nesting level at each step. For example:

```
s(%12(%34))>(%127)>0=0e
```

In this CoT representation:

- The initial expression is `s(%12(%34))`

- The first step resolves the innermost operation: `(%34)` becomes '7'

- The intermediate result is shown: `s(%12(7))>(%127)`

- The process continues until the final result is reached: `s(%12(7))>(%127)>0=0e`

This CoT approach serves multiple purposes: 1. It guides the model through the problem-solving process, mimicking human-like reasoning. 2. It provides more granular supervision, potentially aiding in learning complex operations. 3. It allows us to study how models learn to break down and solve nested problems. Our experiments show that this CoT method significantly improves model performance, particularly for the challenging sum modulo 10 operation (Fig. 7).

## I  OBSERVATIONS FROM THE NORM OF ATTENTION AND FEEDFORWARD OUTPUTS

To investigate the internal dynamics of our models, we focused on the final layer of a 3-layer transformer network, featuring a single attention head and an embedding dimension of 128. Our analysis centered on comparing the behavior of models trained on "All3" operations (max, median, sum) versus those trained solely on the sum operation.

We introduced a novel metric to quantify the impact of different components within the network: the ratio of output to input norms for both the self-attention (SA) and feedforward (FFN) sublayers. Specifically, we computed:

1. Attention ratio: $r_{attn} = \frac{\|SA(LN_1(x))\|}{\|x\|}$

2. Feedforward ratio: $r_{ffwd} = \frac{\|FFN(LN_2(x_1))\|}{\|x_1\|}$

where $LN_1$ and $LN_2$ are layer normalization operations, and $x_1$ is the output of the self-attention sublayer. These ratios provide insight into how much each component modifies its input, serving as a proxy for the component's impact on the overall computation.

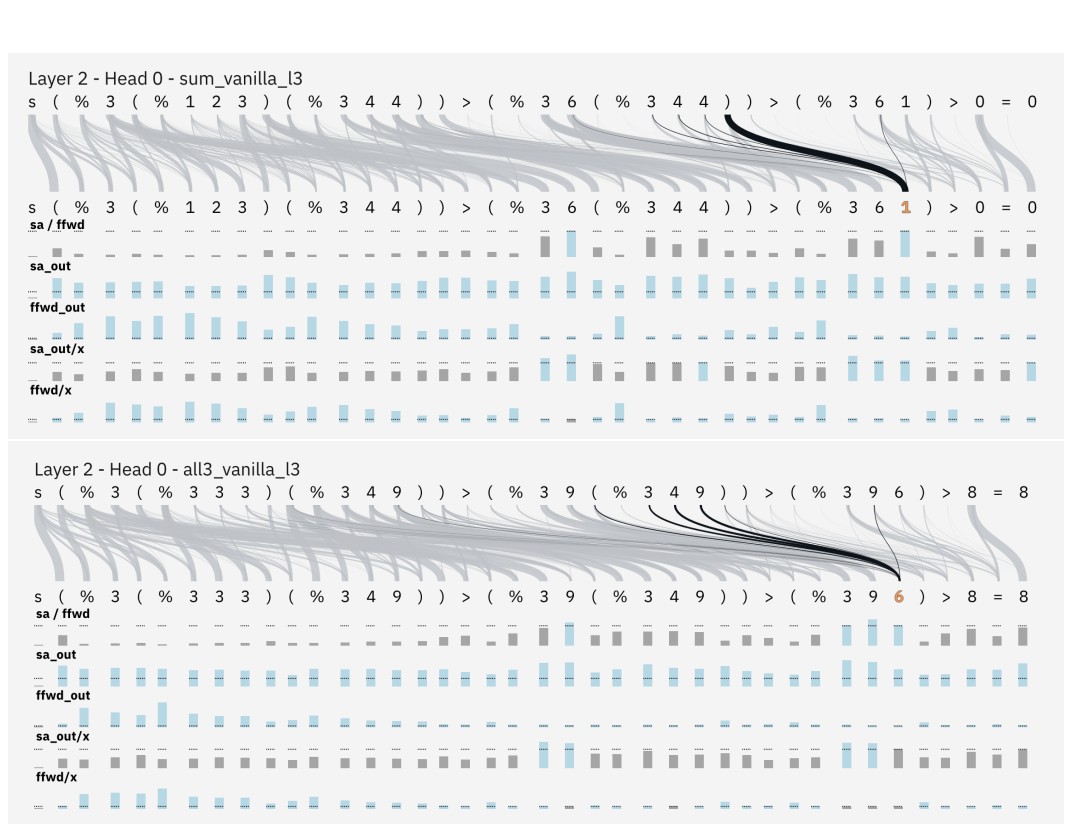

Figure 15: **Attention patterns and layer dynamics in SUM vs ALL3 models.** Each panel shows (CoT) solution to a SUM modulo 10 problem, where '>' indicates solution steps. The first row shows the input sequence, with curved lines representing attention weights from Layer 2 in a 3-layer network. Black lines highlight attention patterns for a specific digit (shown in orange). Below are shown various layer metrics including the ratio of self-attention to feedforward norms (`sa/ffwd`), self-attention output norms (`sa_out`), feedforward output norms (`ffwd_out`), and ratios of layer outputs to inputs (`sa_out/x, ffwd/x`). **Top:** Model trained only on SUM operations shows attention primarily focused on parentheses and structural elements. **Bottom:** Model trained on MAX, MED, and SUM (ALL3) shows attention strongly connecting to digits being combined in each CoT step, suggesting direct involvement in numerical computation. These distinct patterns suggest fundamentally different algorithms learned by each model.

We analyzed the distribution of these ratios across a test set consisting of sum operations for both the 'All3' and 'Sum' models. Kernel Density Estimation (KDE) plots were used to visualize the distributions, and we employed several statistical measures to quantify the differences.

**Attention Sublayer**    The attention sublayer showed moderate but statistically significant differences between the 'All3' and 'Sum' models:

- Kolmogorov-Smirnov test: statistic = 0.1592, p-value < 0.0001

- Jensen-Shannon divergence: 0.1591

- Wasserstein distance: 0.0696

- Effect size (Cohen's d): 0.1761

- 95% CI for mean difference: (0.0466, 0.0860)

The KDE plot revealed that the 'All3' model's attention ratio distribution was more concentrated and peaked higher than the 'Sum' model's distribution. The positive effect size and confidence interval indicate that the 'All3' model generally had higher attention ratios.

**Feedforward Sublayer**    The feedforward sublayer exhibited more pronounced differences:

- Kolmogorov-Smirnov test: statistic = 0.2461, p-value < 0.0001

- Jensen-Shannon divergence: 0.1617

- Wasserstein distance: 0.2830

- Effect size (Cohen's d): -0.3379

- 95% CI for mean difference: (-0.3042, -0.2226)

The KDE plot for the feedforward ratios showed a clear shift between the two distributions. The 'Sum' model's distribution was shifted towards higher values, as confirmed by the negative effect size and confidence interval.

**Interpretation**    These results reveal distinct operational patterns between models trained on 'All3' operations versus those trained solely on 'Sum':

1. In the attention sublayer, the 'All3' model shows slightly higher ratio values, **suggesting that attention mechanisms play a more pronounced role when the model is trained on diverse operations.** This could indicate that the attention sublayer is capturing more complex patterns or relationships necessary for handling multiple operations.

2. Conversely, in the feedforward sublayer, the 'Sum' model demonstrates significantly higher ratio values. This **suggests that when trained on 'Sum' alone, the model relies more heavily on the feedforward network for computation.** This could imply that the 'Sum' operation is being implemented more directly through feedforward transformations.

3. The larger effect size in the feedforward layer (-0.3379) compared to the attention layer (0.1761) indicates that the difference in behavior is more pronounced in the feedforward component.

These observations suggest a trade-off in how the network allocates its computational resources. The 'All3' model appears to leverage its attention mechanism more, potentially to handle the diversity of operations it was trained on. In contrast, the 'Sum' model seems to channel more of its computation through the feedforward network, possibly developing a more specialized but less flexible approach to solving the sum operation.

This analysis provides evidence that the internal dynamics of transformer models adapt significantly based on the diversity of tasks they are trained on, even when evaluated on the same type of operation ('Sum'). It highlights the importance of considering task diversity in understanding and optimizing neural network architectures.

## J   DISCUSSION

**Implications for Algorithm Learning and Model Efficiency**   These findings have several important implications:

1. **Efficient Algorithm Discovery:** The success of the hybrid training approach suggests that in the vast search space of possible algorithms, number-based algorithms for SUM may be challenging to discover without an informative representation of numbers. Training on MAX and MED appears to constrain this search space by fostering the development of crucial number properties in the embedding space.

2. **Analytical vs. Memorization-based Learning:** The stark difference in model size and learning dynamics between the hybrid-trained and vanilla SUM models suggests that the former learns a more analytical, generalizable approach, while the latter may rely more heavily on memorization.

3. **Representation Transfer:** The retention of number-like embeddings even after MAX and MED are no longer trained on demonstrates the stability and transferability of learned representations.

4. **Multi-task Synergy:** The rapid, near-simultaneous learning of multiple operations in the All3 model underscores the synergistic benefits of multi-task learning in developing rich, generalizable internal representations.

**Implications and Ramifications**

1. **Algorithm Inference:** Our findings suggest that the structure of the embedding space, as captured by the cosine correlation matrix, can serve as a powerful tool for inferring the type of algorithm a model has learned.

2. **Task Complexity Estimation:** The revised KC calculation method, which takes into account the model's potential numerical understanding, provides a more accurate measure of task complexity.

3. **Unified Scaling Law:** The emergence of a consistent power-law relationship between KC and transition point across all task combinations suggests a fundamental principle in how neural networks scale with task complexity.

4. **Multi-task Synergy:** The fact that models trained on multiple operations often outperform those trained on individual tasks (particularly SUM) underscores the potential benefits of multi-task learning in developing more efficient and generalizable representations.

5. **Rethinking Complexity Measures:** Our results highlight the importance of considering the learned representations when estimating task complexity.

6. **Insights into Model Capacity:** The power-law relationship provides a quantitative framework for understanding how model capacity relates to task complexity, potentially guiding more efficient model design and training strategies.

These findings not only provide a more coherent understanding of the relationship between task complexity and model learning in our ListOps experiments but also open up new avenues for research in neural network scaling laws, multi-task learning, and the development of more interpretable AI systems. Future work could explore whether similar principles hold across other domains and model architectures, potentially leading to more general theories of neural network learning and capacity.

To further validate these insights and assess the degree to which smaller, hybrid-trained models employ more "analytical" approaches compared to larger, single-task models, we conducted additional probing experiments. These involved evaluating models on holdout datasets and analyzing the distribution of outputs from their attention and feed-forward modules, which we will explore in detail in the following section.

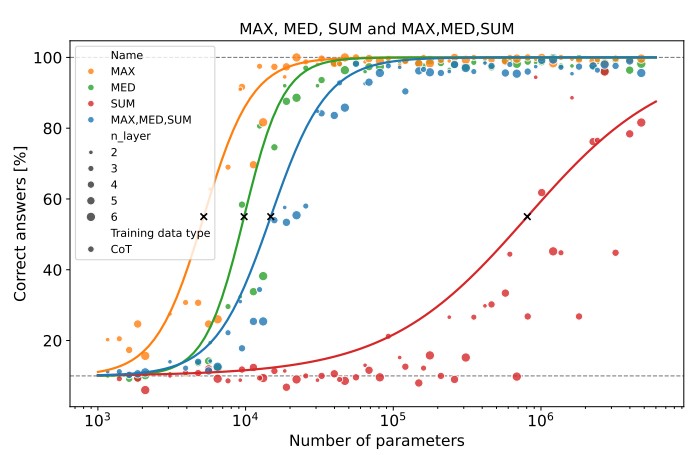

Figure 16: **Operations.**

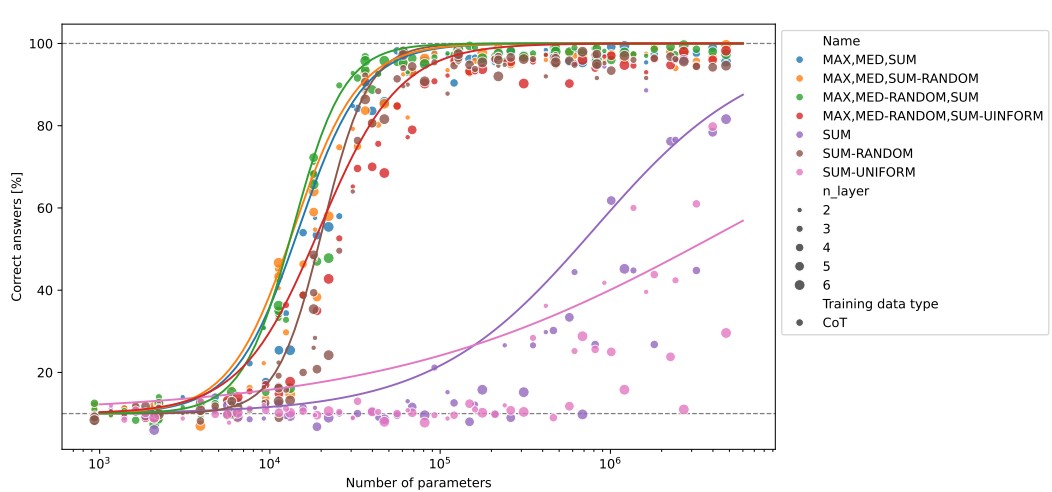

Figure 17: **Random sum table.**

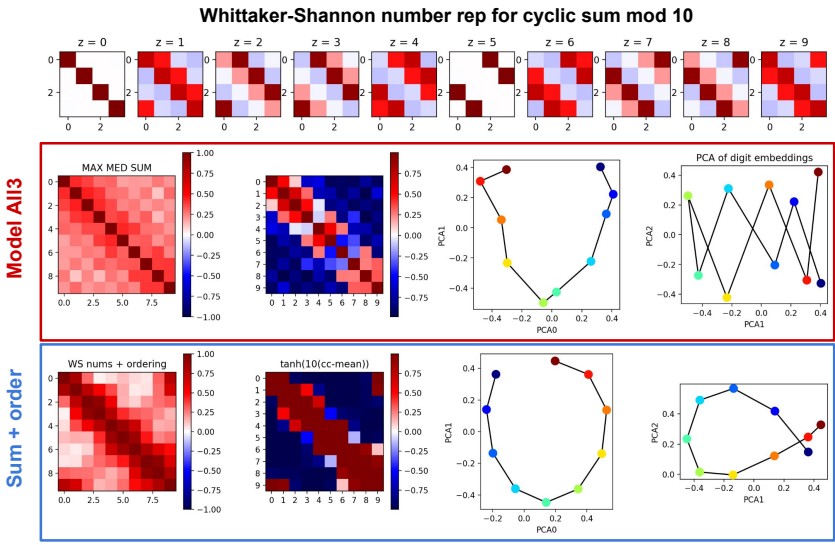

Figure 18: Evidence for numbers.

## K  ABLATION STUDIES

## L  MIXING COMPLEXITY

**Where does the complexity come from?** There are multiple metrics to determine the complexity of a LisOps equation such as the number of characters ($C_{Character}$), the nesting level ($C_{Nesting}$), the number of operands, the number of branching ($C_{Branching}$), operation diversity ($C_{Diversity}$), or the type operation itself ($C_{OperationComplexity}$). Here we introduce a metric that takes the weighted sum of all these metrics with equal weight ($w = 0.2$):

$$C_{total} = wC_{Character} + wC_{Nesting} + wC_{Branching} + wC_{Diversity} + wC_{OperationComplexity} \quad (1)$$

We calculate the operation complexity ($C_{OperationComplexity}$) based on the Kolmogorov complexity of each operation ($KC.$) separately and weighted with the number of operations ($N.$):

$$C_{OperationComplexity} = N_- KC_- + N_+ KC_+ + N_/ KC_/ + N_\% KC_\% \quad (2)$$

The Kolmogorov complexity is defined as the length of the shortest code or algorithm capable of generating a given output. In Python, we approximate this complexity by implementing each operation individually, using a parser to solve the equation without relying on built-in Python functions, and then measuring the length of the compressed file. Assuming that the model recognizes numbers as symbols we implement the max, min, and med operations using a sorted list of symbols e.g. $0 < 1 < 2 < 3 < \ldots$ and the sum module 10 operation using a character table e.g. $[[(0,0),0],[(0,1),1],\ldots[(9,9),8]]$.

**When does the model learn?** We analyze the learning capabilities of a small GPT-style model, which is based on the transformer architecture Vaswani et al. (2017); Karpathy, utilizing the ListOps dataset (maximum 3 nesting levels and 3 arguments). The model is built on a single-layer transformer block paired with an MLP layer, allowing us to control the number of trainable parameters by tuning the embedding dimension (from 4 to 256) and the number of layers (from 1 to 6). We train each model on all possible operation combinations and measure their performance in terms of accuracy, observing that models with a critical number of trainable parameters ($n_{critical}$) successfully learn the task (Fig. 19). $n_{critical}$ highly depends on the operation combinations, meaning the task complexity, is low for simple operations such as min or max, and high for more complex operations like sum. Surprisingly, we observe that combining the sum task with any other operation(s) in the training data makes it easier to learn for the model. Also, we find that applying our extensive complexity metric ($C_{total}$) 2 on the training data and calculating the average complexity for each operation combination the sum task has the highest complexity and the diversified task are less complex showing a strong correlation between complexity and the critical number of parameters required for successful learning (Fig. 19b).

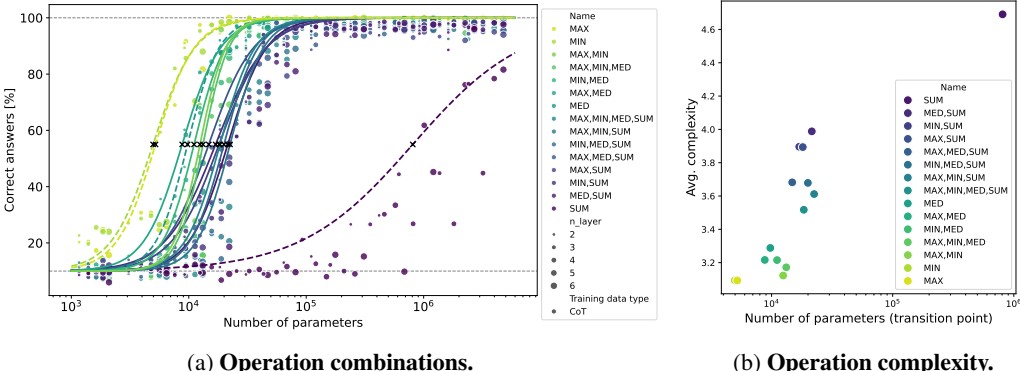

(a) **Operation combinations.**    (b) **Operation complexity.**

Figure 19

**What does the model learn? Number or not number?** This observation raises the question: What exactly is the model learning?

Our initial hypothesis suggests that by combining the max, median, and sum operations, the model may learn some underlying rules about the symbols, effectively recognizing them as numbers, thereby

simplifying the calculations. To explore this hypothesis, we examine the embeddings of the trained model and calculate the cosine similarity between each character's embedding. Notably, in the cases of the max and median operations, the cosine similarity matrix reveals a strong correlation between sequential numbers, indicating that the model has indeed learned the correct numerical order. While the cosine similarity matrix for the sum operation alone does not exhibit this correlation, combining operations results in a strong correlation, suggesting that diversifying the task enhances the model's understanding of numerical relationships.

Our secondary hypothesis is that if no inherent numerical relationships exist between tasks, the learning process should become more challenging, requiring a higher number of parameters for the model to learn the tasks. To falsify or strengthen the number hypothesis we generate randomly sorted symbol lists for the max and median calculations, as well as a randomly generated character table for the sum modulo 10 operation.

## L.1 CODE COMPLEXITY

Listing 1: MAX,MIN,MED,SUM symbolic algorithm based on lookup table

```
1  d = ['0', '1', '2', '3', '4', '5', '6', '7', '8', '9']
2
3  u = [[('0', '0'), '0'], [('0', '1'), '1'], [('0', '2'), '2'],[('0', '3'),
       '3'], [('0', '4'), '4'],
4   [('0', '5'), '5'], [('0', '6'), '6'], [('0', '7'), '7'], [('0', '8'), '8
       '], [('0', '9'), '9'], [('1', '0'), '1'],
5   [('1', '1'), '2'], [('1', '2'), '3'], [('1', '3'), '4'], [('1', '4'), '5
       '], [('1', '5'), '6'], [('1', '6'), '7'],
6   [('1', '7'), '8'], [('1', '8'), '9'], [('1', '9'), '0'], [('2', '0'), '2
       '], [('2', '1'), '3'], [('2', '2'), '4'],
7   [('2', '3'), '5'], [('2', '4'), '6'], [('2', '5'), '7'], [('2', '6'), '8
       '], [('2', '7'), '9'], [('2', '8'), '0'],
8   [('2', '9'), '1'], [('3', '0'), '3'], [('3', '1'), '4'], [('3', '2'), '5
       '], [('3', '3'), '6'], [('3', '4'), '7'],
9   [('3', '5'), '8'], [('3', '6'), '9'], [('3', '7'), '0'], [('3', '8'), '1
       '], [('3', '9'), '2'], [('4', '0'), '4'],
10  [('4', '1'), '5'], [('4', '2'), '6'], [('4', '3'), '7'], [('4', '4'), '8
       '], [('4', '5'), '9'], [('4', '6'), '0'],
11  [('4', '7'), '1'], [('4', '8'), '2'], [('4', '9'), '3'], [('5', '0'), '5
       '], [('5', '1'), '6'], [('5', '2'), '7'],
12  [('5', '3'), '8'], [('5', '4'), '9'], [('5', '5'), '0'], [('5', '6'), '1
       '], [('5', '7'), '2'], [('5', '8'), '3'],
13  [('5', '9'), '4'], [('6', '0'), '6'], [('6', '1'), '7'], [('6', '2'), '8
       '], [('6', '3'), '9'], [('6', '4'), '0'],
14  [('6', '5'), '1'], [('6', '6'), '2'], [('6', '7'), '3'], [('6', '8'), '4
       '], [('6', '9'), '5'], [('7', '0'), '7'],
15  [('7', '1'), '8'], [('7', '2'), '9'], [('7', '3'), '0'], [('7', '4'), '1
       '], [('7', '5'), '2'], [('7', '6'), '3'],
16  [('7', '7'), '4'], [('7', '8'), '5'], [('7', '9'), '6'], [('8', '0'), '8
       '], [('8', '1'), '9'], [('8', '2'), '0'],
17  [('8', '3'), '1'], [('8', '4'), '2'], [('8', '5'), '3'], [('8', '6'), '4
       '], [('8', '7'), '5'], [('8', '8'), '6'],
18  [('8', '9'), '7'], [('9', '0'), '9'], [('9', '1'), '0'], [('9', '2'), '1
       '], [('9', '3'), '2'], [('9', '4'), '3'],
19  [('9', '5'), '4'], [('9', '6'), '5'], [('9', '7'), '6'], [('9', '8'), '7
       '], [('9', '9'), '8']]
20
21  def ind(l, u):
22      for i in range(len(u)):
23          if u[i][0] == l:
24              return i
25      return -1
26
27  def sum(l, u):
28      if len(l) == 1:
29          return l[0]
```

```python
    else:
        s_ = u[ind((l[0], l[1]), u)][1]
        for i in l[2:]:
            s_ = u[ind((s_, i), u)][1]
        return s_

def max(l, d):
    m_ = l[0]
    for i in l:
        if d.index(i) > d.index(m_):
            m_ = i
    return m_

def sort(l, d):
    for i in range(len(l)):
        for j in range(i+1, len(l)):
            if d.index(l[i]) > d.index(l[j]):
                s = l[i]
                l[i] = l[j]
                l[j] = s
    return l

def med(l, d):
    l = sort(l, d)
    n = len(l)
    return l[n//2]

def solve(e, ordered_min=d, ordered_max=d, ordered_median=d, sum_table=u)
    :
    open_parenthesis = []
    for en, i in enumerate(e):
        if i == '(':
            open_parenthesis.append(en)
        if i == ')':
            start = open_parenthesis.pop()

            elements = [j for j in e[start+2:en]]
            if e[start+1] == '+':
                e = e[:start] + str(max(elements, ordered_max)) + e[en
                    +1:]
                break
            elif e[start+1] == '%':
                e = e[:start] + str(sum(elements, sum_table)) + e[en+1:]
                break
            elif e[start+1] == '/':
                e = e[:start] + str(med(elements, ordered_median)) + e[en
                    +1:]
                break

    return solve(e, ordered_min, ordered_max, ordered_median, sum_table)
        if '(' in e else e
```

Listing 2: MAX,MIN,MED,SUM number based algorithm - binary opeartion

```python
d = ['0', '1', '2', '3', '4', '5', '6', '7', '8', '9']

def ba(a, b):
    while b != 0:
        carry = a & b
        a = a ^ b
        b = carry << 1
    return a
```

```python
def bs(a, b):
    while b != 0:
        borrow = (~a) & b
        a = a ^ b
        b = borrow << 1
    return a

def s(a, b):
    sr = ba(a, b)
    while sr >= 10:
        sr = bs(sr, 10)
    return sr

def sum(l, d):
    if len(l) == 1:
        return l[0]
    else:
        s_ = s(d.index(l[0]),d.index(l[1]))
        for i in l[2:]:
            s_ = s(d.index(s_) + d.index(i))
        return s_

def max(l, d):
    m_ = l[0]
    for i in l:
        if d.index(i) > d.index(m_):
            m_ = i
    return m_

def min(l, d):
    m_ = l[0]
    for i in l:
        if d.index(i) < d.index(m_):
            m_ = i
    return m_

def sort(l, d):
    for i in range(len(l)):
        for j in range(i+1, len(l)):
            if d.index(l[i]) > d.index(l[j]):
                s = l[i]
                l[i] = l[j]
                l[j] = s
    return l

def med(l, d):
    l = sort(l, d)
    n = len(l)
    return l[n//2]

def solve(e, ordered_min=d, ordered_max=d, ordered_med=d, sum_table=d):
    open_parenthesis = []
    for en, i in enumerate(e):
        if i == '(':
            open_parenthesis.append(en)
        if i == ')':
            start = open_parenthesis.pop()

            elements = [j for j in e[start+2:en]]
            if e[start+1] == '+':
                e = e[:start] + str(max(elements, ordered_max)) + e[en
                    +1:]
                break
```

```
            elif e[start+1] == '-':
                e = e[:start] + str(min(elements, ordered_min)) + e[en
                    +1:]
                break
            elif e[start+1] == '%':
                e = e[:start] + str(sum(elements, sum_table)) + e[en+1:]
                break
            elif e[start+1] == '/':
                e = e[:start] + str(med(elements, ordered_med)) + e[en
                    +1:]
                break

    return solve(e, ordered_min, ordered_max, ordered_med, sum_table) if
        '(' in e else e
```

