# OpenReview forum: "The Role of Task Complexity in Emergent Abilities of Small Language Models"
_ICLR.cc/2025/Conference — ICLR 2025 Conference Withdrawn Submission_

### Official Review · Reviewer_Fr5d · 2024-10-22

**Soundness:** 2
**Presentation:** 3
**Contribution:** 2
**Rating:** 3
**Confidence:** 4

**Summary:**

This paper investigates how the complexity of tasks affects the performance and parameter requirements of small transformer models. It focuses on the relationship between task complexity and model size, using the Kolmogorov complexity (a measure of how difficult it is to describe a task) to estimate the difficulty of various tasks from the ListOps dataset. They find that there is a direct power-law relationship between complexity and the number of parameters needed to learn the task. They also demonstrate that learning tasks of different complexities together is helpful for learning more robust representations.

**Strengths:**

I think the presentation of the work is well done and the paper is quite well focused on the specific setting that is motivated in the introduction. The paper does a fair job at attempting to answer the problem they describe.

**Weaknesses:**

One concern here is that the results are purely empirical. The relationship between the number of parameters necessary to solve a task and the Kolmogorov complexity is rather weak based on what the results demonstrate, in particular as Figure 2 actually demonstrates quite a few tasks to clutter around the same number of parameters.

Additionally, the point that joint training improves task performance is not very surprising, as the model must learn to adapt to different settings rather than a single one and therefore the fact that the embeddings express some more meaningful embeddings is hardly a new discovery.

This leads to my largest concern which is that the main claim of the paper (that jointly training on different tasks of varying complexity and diversity leads to more robust models/representations) is something that has been explored rather frequently at nauseam, in particular with respect to language models [1, 2, 3]. This, combined with the fact that the authors only provide fairly incomplete justifications of how Kolmogorov complexity affects the ability to learn tasks leaves this rather lacking and I'm not convinced that the arguments made for such as specific setting such as ListOps is exactly relevant to general use cases of language models.

-----

[1]  Xie et al., Doremi: Optimizing data mixtures speeds up language model pretraining. NeurIPS 2023.

[2]  Xie et al., Data Selection for Language Models via Importance Resampling. NeurIPS 2023.

[3] Miranda et al., Beyond Scale: The Diversity Coefficient as a Data Quality Metric for Variability in Natural Language Data. arXiv, 2023.

**Questions:**

- Assuming that Kolmogorov complexity is not the only way of measuring task complexity, have the authors tried to actually view these from alternative complexity measures (ex. as decision problems in formal language).
- As a more controlled experiment, it might be nice to actually observe these learning dynamics with a simpler Transformer model, for example one where the vocabulary is explicitly restricted to the relevant symbols in ListOps.
- The actual length of the example affects the difficulty of the task since it can often require the model to be able to parse through a larger structure, which can be difficult based on the context-window size of the model. I believe this is something that should be considered in the experimental results.

---

### Official Review · Reviewer_JbRh · 2024-10-29

**Soundness:** 1
**Presentation:** 1
**Contribution:** 2
**Rating:** 3
**Confidence:** 4

**Summary:**

The paper studied the "emergence" capabilities on small arithmetic tasks on increasing size of language models. The paper investigated the relationship between the task complexity (characterized by the Kolmogorov Complexity) and the point of "emergence" - i.e. sudden increase of accuracy, and fit a "scaling law" between them.

**Strengths:**

1. Using gzip and Kolmogorov Complexity to characterize task complexity.
2. Thoroughly studied the embedding layer weights, and found circular patterns in them when plotting the cosine correlations between tokens.

**Weaknesses:**

1. Similar approaches have been proposed by others, such as https://arxiv.org/pdf/2405.16684 -> also using gzip to characterize task complexity

2. Proposed task complexity measure only works for ListOps. It is not scalable or generalizable. For example, how do you define the "minimal python implementations" for a large python codebase with multiple files?

3. In Figure 1, right panel, you plotted a line fitting only 3 algorithms. 3 algorithms is too small for any kind of curve fitting that indicates any scaling laws.

4. You characterized the algorithms that the neural network could develop to solve MAX, MED, MIN, and SUM as only 2 categories: symbolic and numeric. However, there is no guarantee that the neural network could discover some new algorithms that does not fit in your categories, such as a mix of two.  The way to figure out what the neural network actually learned is to find a circuit systematically, with the approach like [this paper](https://openreview.net/forum?id=NpsVSN6o4ul)

5. Proposed task complexity measure has 2 parts "symbolic KC" and "numerical KC". The measure is not consistent, and also due to the points above, not generalizable to other tasks.

6. In section 3.2, first paragraph, you assumed "MAX+MIN+MED" has a higher task complexity than "SUM". However, there is no proof showing that the combination of these tasks has a higher task complexity.

7. In section 3.3, you only inspected the learned embeddings, then concluded that the learned "SUM" algorithm "has not developed a structured representation of numbers". However, you should inspect other parts of the model before concluding that. Similarly, there is no evidence concluding that the "SUM" algorithm memorizes the numeric table.

**Questions:**

1. There are multiple occurrences of [insert number] in the paper.
2. Some parts are incomplete, like B. Designing solution algorithms, and K. Ablation Studies section in the appendix.

---

### Official Review · Reviewer_fu55 · 2024-11-03

**Soundness:** 3
**Presentation:** 3
**Contribution:** 2
**Rating:** 3
**Confidence:** 3

**Summary:**

The paper investigates the relationship between task complexity and the minimum model size required for learning specific tasks in small transformer models. It focuses on the ListOps dataset, which consists of nested mathematical operations, and defines task complexity using a proxy for Kolmogorov complexity (KC) based on the compressed size of minimal Python code implementing the tasks.

**Strengths:**

1. The paper offers a fresh perspective on how task complexity influences the scaling laws of language models, which traditionally have been considered largely independent of the tasks being learned.
2. Kolmogorov complexity is connected to the scaling law, which is a new perspective
3. The paper also analysis the embedding learned and justify the emergence of numeric representation.
4. The process of deriving approximate Kolmogorov complexity is rigorous.

**Weaknesses:**

1. Although the connection between Kolmogorov complexity and the emergence point is novel, the underlying mechanism in the other part of the paper appears to lack significant novelty. In other words, it does not provide substantial additional information to our current understanding of large language models (LLMs). For instance, it is evident that the SUM-only operation will not have a numeric representation as the basis of this operation, regardless of how many times it is composed, is to map 100 classes to ten classes. It is considerably easier to memorize than to understand. Given that the model must first have an understanding of operation composition, the remaining task is to deterministically map 100 instances to ten classes. Therefore, it is indeed far easier to memorize than to understand. Regarding the max, min, and median operations, for example, if the length of the array is 3, it will map 1000 to 10 classes, making it easier to understand than to memorize.

2. Kolmogorov complexity is extremely difficult to compute for any practical tasks. Consequently, the practical implications of this scaling law are limited.

3. There appears to be a substantial amount of content in the appendix. However, the current version of the main body contains verbose sentences and large spaces. Some of the appendix content should be incorporated into the main content to make the experiment setting more explicit.

4. It is contended that the support for the claim "multitask learning accelerates the learning of more challenging tasks" is not adequately supported by the experiments. Since SUMUP is accelerated by MAX, MIN, and MED only because the latter tasks assist in learning a numeric representation. Multitask learning in this context is merely a confounding factor. To support this claim, experiments should be conducted that demonstrate something like: MAX only does not help MIN, MED, or SUMUP, but MAX and MIN together can help MED and SUMUP.

5. I suspect some sentences are written by LLMs, or are polished by LLMs. Although it is not a big issue, the author should make such sentences more natural. E.g., “As we push the boundaries of AI capabilities, understanding these nuanced relationships between task complexity...”

**Questions:**

See weakness

---

### Official Review · Reviewer_ceFf · 2024-11-05

**Soundness:** 3
**Presentation:** 2
**Contribution:** 2
**Rating:** 3
**Confidence:** 4

**Summary:**

The paper studies the relationship between task complexity (measured through a proxy for Kolmogorov complexity) and the minimum model parameter size required for learning simple arithmetic tasks in small transformer models. They find that the number of parameters required to learn a task increases as a cubic function in task complexity. They find that SUM mod 10 is harder to learn from MAX, MEDIAN and interestingly, it is easier to learn alongside the other two. They also perform analysis on the model embeddings to infer the kind of algorithm that the model is learning.

**Strengths:**

1. The paper makes an attempt at understanding the learning mechanism behind how small transformers acquire arithmetic abilities - which is currently an interesting area of study. It also makes an attempt at doing so rigorously using the concept of Kolmogorov Complexity.
2. I find the analysis of the structure of the learned embeddings through cosine similarity and PCA to be particularly interesting. The evidence that the model learns how to sort the different numbers and picks up parity on the way to learning SUM mod 10 is an important step towards better understanding the algorithms that transformers are learning.

**Weaknesses:**

1. While the Kolmogorov complexity is extremely interesting to study, using compressed size of minimal python code is somewhat problematic as a proxy, particularly for simple arithmetic operations. I appreciate the authors listing out the limitations of this approach and discussing its precedent but the rules specified in Appendic C seem somewhat arbitrary and I expect the compressed sizes of the 3 algorithms to be fairly similar.
2. The accuracy definition which only tests the first character after the first `=` symbol seems fairly forgiving. Do the authors have a measure of what the "strict" accuracy would be? This is assuming that the model is expected to also output `e` - the end token.
3. I would appreciate if the authors edited the plots to make them easier to read/parse. The font sizes are far too small and the legends are fairly difficult to parse.
4. The fitted curve that results in the cubic scaling law (KC $\approx 18n_p^{0.34}$) seems to be based on too few datapoints to rely on.



Typos:
1. SUM vs $'$sum$'$. I recommend the authors pick a standard notation and stick to it.
2. Missing parenthesis after Appendix I on page 10.
3. [insert number]x on page 10.
4. Page 3 line 135: "representing the "$\rightarrow$ "representing them"

**Questions:**

1. In Page 5, the authors state that the naive solution for max is unlikely to be learned since it requires large memory - isn't this just a 10x10 table for ListOps?
2. Can the authors provide some intuition for the attention output/input ratio as a metric in the end of section 3.5? I also recommend the authors define it before referring to "higher ratio values".
3. Doesn't using for loops in Appendix C assume comparisons even though we are trying to avoid them for symbolic algorithms?
4. Appendix C.1 seems to be missing.
5. In the default setting where number of operands and maximum nesting is 3, isn't the total number of possible equations around 50k? I am perhaps misunderstanding what the most complex equation for a given nesting, num_operands is. I am trying to understand what fraction of all possible examples the model is seeing during training. Also, it seems like 1k samples is too small a test set. I would like to see if the accuracy over the test set is representative of the accuracy over the full universe of examples up to a maximum nesting, num_operands.

---

### Note · Authors · 2024-11-27

**Comment:**

We would like to thank the reviewers for their valuable feedback. It is clear that more work needs to be done, so we have decided to withdraw.
We would like to also briefly comment on some of the main points raised by the reviewers:
- **use of gzip not new:** To our knowledge, all past works have focused on dataset complexity, captured by zipped size. The key novelty of our work is the use **Kolmogorov Complexity**, which is *not* the dataset compression, but rather the size of the *algorithm* generating the solution.
- **Kolmogorov Complexity as task complexity:** We believe KC is a more accurate measure of task complexity than the data entropy used in past works, and that KC touches on something much deeper. For example, consider two datasets: 1) an arithmetic series mod 10; 2) a pseudo-random series. It may be that the code generating both series has similar KC, but the compressed size of 1 should be much smaller than 2. The question we wanted to tackle was, which of the two measures determines minimum model size to solve the task.

- **KC analysis not scalable:** Thank you, this is a very important point, but we disagree that it cannot be applied to anything other than ListOps. An important example where we can apply it is code: Most production software contains procedurally generated or compiled code, where a small source code generates a large amount of production code. The KC analysis can be applied to any dataset for which we have a generating code. This includes many mathematical series, any procedurally generated data, and many coding datasets. We agree that in its current form our analysis cannot be applied to natural language. But we believe that shouldn’t prohibit us from observing potentially deep connections between model size and code complexity.
- **Too few data points for scaling:** Note that each point in our KC vs number of parameters plots is a **transition point**, which is the results of about a thousand training runs. Hence the dozen points on the plot required over 10 thousand runs and they consist of all possible combinations of the four operations in ListOps.
- **Not a proof that the model learned the code we used:** This is perhaps the most challenging issue with our current paper and we acknowledge that we do not have a definitive proof. We thank the reviewer JbRh and others for pointing us to papers trying to identify circuits doing specific tasks. We will work on this further. We do have some more evidence showing that the algorithm used by ALL3 is fundamentally different from SUM alone (new Fig. 15 in revised paper). It shows the attention module in ALL3 attending to numbers during CoT step, while in SUM attention is on the parenthesis. The following section details what we learned about this difference in algorithms.

We plan to continue this work and hope to get more conclusive and convincing results. Thank you all.

## More evidence on algorithms learned by SUM vs ALL3 (joint training)

We appreciate the reviewer's concern about algorithmic differences. While we agree that direct proof of memorization vs. arithmetic understanding is challenging, we have multiple lines of evidence suggesting fundamentally different algorithms are learned:

1. Embedding Analysis:
- SUM-only models show no numerical properties in embeddings
- Joint training (ALL3) develops clear numerical ordering and even discovers number parity

2. Attention Pattern Analysis (new evidence):
- In SUM-only models, attention primarily focuses on parentheses and nesting structure, suggesting the model tracks syntactic structure rather than numerical relationships
- In ALL3 models, attention weights strongly connect to the actual digits being combined in each Chain-of-Thought step, indicating direct involvement in numerical computation
- This distinct attention behavior suggests ALL3 models actively use attention for arithmetic operations, while SUM-only models use it primarily for parsing

3. Layer Dynamics:
- Different ratios of self-attention vs. feedforward activity between the two models
- SUM-only models rely more heavily on feedforward layers, consistent with lookup-table behavior
- ALL3 models show more balanced use of attention and feedforward mechanisms, suggesting more sophisticated arithmetic processing

While we cannot definitively prove memorization vs. arithmetic understanding, these multiple independent indicators strongly suggest different algorithmic approaches. The SUM-only model's behavior is consistent with memorization (no number properties in embeddings, attention focused on syntax rather than digits, heavy reliance on feedforward layers), while the ALL3 model shows signs of numerical understanding (structured embeddings, arithmetic-focused attention patterns, balanced layer utilization).

**Withdrawal Confirmation:**

I have read and agree with the venue's withdrawal policy on behalf of myself and my co-authors.